# MicroRNA-like snoRNA-Derived RNAs (sdRNAs) Promote Castration-Resistant Prostate Cancer

**DOI:** 10.3390/cells11081302

**Published:** 2022-04-12

**Authors:** Alexander B. Coley, Ashlyn N. Stahly, Mohan V. Kasukurthi, Addison A. Barchie, Sam B. Hutcheson, Dominika Houserova, Yulong Huang, Brianna C. Watters, Valeria M. King, Meghan A. Dean, Justin T. Roberts, Jeffrey D. DeMeis, Krisha V. Amin, Cameron H. McInnis, Noel L. Godang, Ryan M. Wright, David F. Haider, Neha B. Piracha, Cana L. Brown, Zohaib M. Ijaz, Shengyu Li, Yaguang Xi, Oliver G. McDonald, Jingshan Huang, Glen M. Borchert

**Affiliations:** 1Department of Pharmacology, College of Medicine, University of South Alabama, Mobile, AL 36608, USA; abc1323@jagmail.southalabama.edu (A.B.C.); aab1623@jagmail.southalabama.edu (A.A.B.); sbh1821@jagmail.southalabama.edu (S.B.H.); dh1001@jagmail.southalabama.edu (D.H.); yh1623@jagmail.southalabama.edu (Y.H.); bcw1622@jagmail.southalabama.edu (B.C.W.); mad1521@jagmail.southalabama.edu (M.A.D.); justin.roberts@cuanschutz.edu (J.T.R.); jdd1527@jagmail.southalabama.edu (J.D.D.); kva1521@jagmail.southalabama.edu (K.V.A.); chm1522@jagmail.southalabama.edu (C.H.M.); ng1823@jagmail.southalabama.edu (N.L.G.); rmw1721@jagmail.southalabama.edu (R.M.W.); dfh1821@jagmail.southalabama.edu (D.F.H.); nbp1721@jagmail.southalabama.edu (N.B.P.); clb1626@jagmail.southalabama.edu (C.L.B.); zmi1321@jagmail.southalabama.edu (Z.M.I.); 2Medical Scientist Training Program, University of Colorado School of Medicine, Aurora, CO 80045, USA; ashlyn.stahly@cuanschutz.edu; 3School of Computing, University of South Alabama, Mobile, AL 36608, USA; mk1530@jagmail.southalabama.edu (M.V.K.); sl1721@jagmail.southalabama.edu (S.L.); huang@southalabama.edu (J.H.); 4Department of Biology, University of South Alabama, Mobile, AL 36608, USA; vmk902@berkeley.edu; 5Department of Molecular and Cell Biology, University of California Berkeley, Berkeley, CA 94720, USA; 6Department of Biochemistry and Molecular Genetics, University of Colorado School of Medicine, Aurora, CO 80045, USA; 7Department of Genetics, School of Medicine, Louisiana State University Health Sciences Center, New Orleans, LA 70112, USA; yxi@lsuhsc.edu; 8Stanley S. Scott Cancer Center, Louisiana State University Health Sciences Center, New Orleans, LA 70112, USA; 9Department of Pathology, Sylvester Comprehensive Cancer Center, University of Miami, Miami, FL 33146, USA; ogm443@med.miami.edu

**Keywords:** cancer, prostate, castration-resistant, noncoding RNA, snoRNA, sdRNA, CD44, CDK12, microRNA, ncRNA

## Abstract

We have identified 38 specifically excised, differentially expressed snoRNA fragments (sdRNAs) in TCGA prostate cancer (PCa) patient samples as compared to normal prostate controls. SnoRNA-derived fragments sdRNA-D19b and -A24 emerged among the most differentially expressed and were selected for further experimentation. We found that the overexpression of either sdRNA significantly increased PC3 (a well-established model of castration-resistant prostate cancer (CRPC)) cell proliferation, and that sdRNA-D19b overexpression also markedly increased the rate of PC3 cell migration. In addition, both sdRNAs provided drug-specific resistances with sdRNA-D19b levels correlating with paclitaxel resistance and sdRNA-24A conferring dasatinib resistance. In silico and in vitro analyses revealed that two established PCa tumor suppressor genes, CD44 and CDK12, represent targets for sdRNA-D19b and sdRNA-A24, respectively. This outlines a biologically coherent mechanism by which sdRNAs downregulate tumor suppressors in AR-PCa to enhance proliferative and metastatic capabilities and to encourage chemotherapeutic resistance. Aggressive proliferation, rampant metastasis, and recalcitrance to chemotherapy are core characteristics of CRPC that synergize to produce a pathology that ranks second in cancer-related deaths for men. This study defines sdRNA-D19b and -A24 as contributors to AR-PCa, potentially providing novel biomarkers and therapeutic targets of use in PCa clinical intervention.

## 1. Introduction

Recently, the functional repertoires of even the most established types of noncoding RNAs (e.g., transfer RNAs (tRNA) and small nucleolar RNAs (snoRNA)) have been greatly expanded through defining an array of novel activities carried out by specifically excised fragments [1,2,3,4,5]. In particular, the processing of snoRNAs into sno-derived RNAs (sdRNAs) has garnered increasing attention over the past decade. SnoRNAs have long been thought to primarily function as guides of homology-directed post-transcriptional editing of ribosomal RNAs (rRNAs) and other noncoding (ncRNAs) in the nucleolus, which ensures accurate translation of proteins by the ribosome [6]. In 2008, however, widespread, specific processing of snoRNAs into 16–36 nucleotide (nt) fragments, largely indistinguishable from microRNAs (miRNAs), was first reported [1]. Since then, our lab [7] and others [8,9,10,11,12] have independently confirmed that sdRNAs are processed analogously to and function as miRNAs (Figure 1). Notably, our lab has recently developed a web resource for the identification of noncoding RNA fragments present in small RNA-seq datasets, Short Uncharacterized RNA Fragment Recognition (SURFR), and in agreement with other similar tools [13]; the SURFR analysis of over 13,000 TCGA strongly indicates recurrent, functional human sdRNAs that likely rival miRNAs in number [14,15].

Since the first link between miRNA dysregulation and cancer was identified in 2002, miRNAs have been thoroughly investigated in the context of cancer as master regulators of oncogenes and tumor suppressors [16,17]. With the preponderance of studies implicating miRNAs in virtually all cancer types, aberrant miRNA expression has been rightfully proposed as an emerging hallmark of malignancy [18]. One recent example is miR-31, which targets the RASA1 mRNA in pancreatic cancer. RASA1 deactivates RAS and suppresses RAS/MAPK signaling. MiR-31 relieves this repression and enhances MAPK signaling to significantly enhance cell proliferation and drive pancreatic cancer progression [19]. That said, over the past decade, a growing number of studies suggest that sdRNAs could play a similarly significant role in malignancy. In fact, several miRNAs with well-characterized roles in malignancy have been misannotated and actually represent sdRNAs (Appendix A). As an example, in 2011, miR-605 was shown to regulate p53 tumor in colorectal cancer cells [20]. More recently, the downregulation of miR-605 was also shown to promote the proliferation and invasion of prostate cancer cells [21,22,23]. MiR-605, however, is processed in its entirety from a H/ACA box snoRNA and as such was actually the first sdRNA implicated in cancer [24]. Regardless, many additional sdRNAs have now been suggested to serve regulatory roles in various malignancies [7,11,21,25,26,27,28,29]. For example, in 2017, our lab identified sdRNA-93 as a potent inhibitor of breast cancer cell migration and confirmed the sarcosine metabolism protein PIPOX as a cellular target [7].

Of particular relevance to the work reported herein, the androgen signaling axis is vital to the establishment and growth of prostate cancer (PCa) [30] and broadly divides PCa into two principal classes, each carrying important clinical ramifications: androgen-sensitive PCa and CRPC [31]. Prostate malignancies are often readily treatable with androgen deprivation therapies and chemical or surgical castration strategies, typically resulting in disease remission, that last 2–3 years [32]. Unfortunately, PCa remissions are commonly disrupted by diagnosis with more aggressive and treatment-resistant CRPC [33]. The lack of sustainable CRPC treatment options largely contributes to the fact that PCa ranks second in overall cancer-related deaths for men in the United States [34]. While misregulated sdRNAs have been reported in various PCa models, no sdRNAs thought to specifically contribute to the CRPC phenotype have been reported to date [11,21,25]. As such, the work reported herein focuses on identifying and characterizing sdRNA misexpressions directly involved with CRPC pathogenesis.

## 2. Methods

### 2.1. SURFR Alignment and Data Analysis

All samples were acquired from The Cancer Genome Atlas (TCGA) Research network PRAD dataset and are publicly available at https://www.cancer.gov/tcga (accessed on 14 January 2019). The Short Uncharacterized RNA Fragment Recognition (SURFR) tool [14,15] is a publicly available web-based tool that comprehensively profiles ncRNA-derived RNAs from input RNA-seq data http://salts.soc.southalabama.edu/surfr (accessed on 14 January 2019). SURFR analysis of TCGA PRAD and normal prostate control returned expression in reads per million (RPM) for each sdRNA detected. Rstudio [35] was used to calculate differential expression and rank each sdRNA by cancer prevalence (% of TCGA samples that expressed the sdRNA) and differential expression. Significant results were constricted to those sdRNAs with ≥2× fold change in prostate cancer and were expressed at ≥30 RPM in a minimum of 50% of TCGA PRAD small RNA-seq files. To confirm SURFR findings, small RNA-seq files were obtained for the TCGA PRAD dataset (https://www.cancer.gov/tcga (accessed on 14 January 2019)). Alignments between snoRNAs and reads were obtained via BLAST+ (available at https://blast.ncbi.nlm.nih.gov/Blast.cgi (accessed on 14 January 2019)) using the following parameters: 100% identity, word_size = 6, ungapped, and e-value = 0.001. The frequency of alignments to putative sdRNA loci across each full length snoRNA was calculated by counting reads rigidly defined as ≥20 nts and perfect matches (100% identity). PC3 cell Ago pulldown data were obtained from the NCBI SRA (www.ncbi.nlm.nih.gov/sra/ (accessed on 14 January 2019)) with the identifier SRR2966868. Alignments between sdRNAs and Ago pulldown reads were obtained via BLAST+ using the same parameters as listed above.

### 2.2. Validation of sdRNA Expression via Quantitative RT-PCR

Small RNA was isolated using mirVana miRNA Isolation Kit according to the manufacturer’s instructions. Real-time, quantitative PCR was performed to validate sdRNA expression using the All-in-One miRNA qRT-PCR Kit (GeneCopoeia). Reactions were performed in triplicate in a 96-well plates using 0.2 µM of each custom forward and universal reverse primers provided in the kit and 1.5 µg of total RNA in nuclease-free water. qRT-PCR was conducted on the iQ-5 Real-Time PCR Detection System (Bio-Rad) with the following settings: initial polymerase activation and DNA denaturation at 95 °C for 10 min, followed by 40 cycles of 95 °C for 10 s, 60 °C for 20 s, 72 °C for 15 s. Specificity of amplifications was verified using melting curves. qRT-PCR primers are listed in the Appendix A.

### 2.3. Manipulating sdRNA-D19b and -A24 levels

Antisense oligonucleotides were designed to target sdRNAs and ordered as custom IDT^®^ miRNA Inhibitors from IDT (Integrated DNA Technologies, Coralville, IA, USA). Similarly, sdRNA mimics and scrambled controls were ordered as custom miRIDIAN mimics from Dharmacon (GE Healthcare Dharmacon, Inc., Chicago, IL, USA). Mimic and inhibitor sequences are detailed in Appendix A. Cell migration, proliferation, and invasion assays were then performed to observe the effects of manipulating sdRNA-D19b and -A24 levels. Human PC3 cells (ATCC, CR L-1435) were cultured at 37 °C in 25 cm^2^ vented flasks (Corning, Manassas, VA, USA) with DMEM (Corning) supplemented with 10% fetal bovine serum (Corning) and 1% PenStrep (Corning) in a humidified atmosphere at 5% CO_2_. For transient transfections, the cells were cultured in 12-well plates and grown to 60% confluency before transfection with mimics or inhibitors using Lipofectamine RNAiMAX (Life Technologies, Carlsbad, CA, USA).

### 2.4. Phenotypic Assays

*Proliferation assays*. PC3 cells were first transfected with either 100 nmol/L of RNA mimic, antisense RNA (inhibitor), or negative control using Lipofectamine RNAiMAX (Life Technologies, Carlsbad, CA, USA) according to the manufacturer’s protocol. The cell number was determined by trypan blue staining and manual counting at 24, 36, and 48 h post-transfection. Proliferation was determined as the relative cell number compared with the vehicle-treated (0.1% DMSO) controls (*n* ≥ 8). *Cell migration assays*. Scratch assays were used to assess migration. PC3 cells were transfected with inhibitors or mimics in standard Petri dishes (Corning), as described for examining the cell proliferation, then grown to 100% confluence. A 1 cm-wide zone was scratched across the center of each dish, and then images were taken every 3 h using an EVOS XL Core inverted microscope imaging system to assess the rate of migration (*n* ≥ 3). *Examining chemoresistance*. Following transfection, the cells were incubated for 20 min in 5% CO_2_ at 37 °C, after which they were treated with paclitaxel (5 nM), dasatinib (50 nM), cisplatin (50 µM), or DMSO control. Cell survival was determined by methylene blue staining and manual counting at 0, 6, 12, 18, and 24 h post-transfection. Viability was determined as the relative live cell number compared with vehicle-treated (0.1% DMSO) controls (*n* ≥ 3). *Cell invasion assays*. PC3 transfected cells were used for assessment of invasion using a matrigel invasion chamber kit (BD Bioscience, Sparks, MD, USA). The matrigel-coated plates were rehydrated in warm DMEM serum-free medium for 2 h at 37 °C. After removing the medium, cells were suspended in 500 μL blank medium, and then the 750 μL chemoattractant (medium with 10% fetal bovine serum) was added to the well chamber. Cells were then incubated for 36 h in 5% CO_2_ at 37 °C. For the measurement of invading cells, non-invading cells were removed from the upper surface of the membrane by scraping using cotton swabs and invading cells through the matrigel to the bottom of the insert were fixed with paraformaldehyde and then stained with crystal violet for counting (*n* ≥ 3). Cells were observed and photographed using an EVOS XL Core inverted microscope imaging system. Ten random fields of view for each well were quantified by counting the cells in each field and averaging the results.

### 2.5. Vector Construction

Unless otherwise indicated, PCR amplifications were performed in 40 µL reactions at standard concentrations (1.5 mM MgCl_2_, 0.2 mM dNTP, 1x Biolase PCR buffer, 0.5 U Taq (Bioline USA, Inc., Randolph, MA, USA), 0.5 µM each primer) and using standard cycling parameters (94 °C—3 min, (94 °C—30 s 55 °C—30 s, 72 °C—60 s) × 30 cycles, 72 °C—3 min), then, they were cloned into Topo PCR 2.1 (Invitrogen) and sequenced. Antisense reporters were constructed by the standard PCR with primers containing 5′ Xho-I and 3′ Not-I restriction enzyme sites. Following digestion, amplicons were ligated into the Renilla luciferase 3′UTR of psiCheck2 (Promega, Madison, WI, USA) vector linearized with Xho-I and Not-I. Reporter assays were performed as previously [7,36] described, where the presence of an independently transcribed firefly luciferase in these reporters allowed normalization for transfection efficiency. Primer sequences are detailed in Appendix A.

### 2.6. Luciferase Assays

Human embryonic kidney (HEK293) cell line was obtained from GenLantis (San Diego, CA, USA) and cultured in MEM (Mediatech, Herndon, VA, USA) supplemented with 10% fetal bovine serum (Hyclone, Logan, UT, USA), 25 mg/mL streptomycin, and 25 I.U. penicillin (Mediatech). Cells were cultured in a humidified atmosphere with 5% CO_2_ at 37 °C. For luciferase assays, HEK293 cells were cultured in MEM (10% FBS and 1% PS) in 12-well plates. At 90% confluency, cells were transfected following the Lipofectamine 2000 (Invitrogen, Carlsbad, CA, USA) protocol. At 36 h post transfection, cells were scraped from well bottoms and transferred to 1.5 mL Eppendorf tubes. Eppendorfs were centrifuged at 2000 RCF for 3 min, followed by supernatant aspiration and cell resuspension in 300 µL of PBS. Cells were lysed by freeze thaws and debris removed by centrifuging at 3000 RCF for 3 min. A total of 50 µL of supernatant was transferred to a 96-well MicroLite plate (MTX Lab Systems, Vienna, VA, USA), then, firefly and Renilla luciferase activities were measured using the Dual-glo Luciferase^®^ Reporter System (Promega) and a 96-well plate luminometer (Dynex, Worthing, West Sussex, UK). RLUs were calculated as the quotient of Renilla/firefly RLU and normalized to mock.

### 2.7. Statistical Analyses

*Cell proliferation and migration assays*. Treatment effects were assessed using a two-tailed Student’s *t*-test at each time point measurement. To assess the longitudinal effects of treatment, a mixed model was utilized to examine the difference across all groups and between each pair of groups for the whole study period. Data were presented as mean  ±  SD from no less than three independent experiments, and a *p* value < 0.05 was considered significant. For imaging, five microscopic fields randomly chosen from each assay were counted individually, then, the results were averaged. *Luciferase assays*. Data are presented as the average intensity ± standard deviation in four independent experiments. *Quantitative RT-PCR*. Gene expression was calculated via the Delta–Delta cycle threshold method and qRT-PCR data were analyzed by Fisher’s exact test.

## 3. Results

### 3.1. In Silico Identification of PCa-Overexpressed sdRNAs

Our lab has recently developed a web resource to identify and quantify noncoding RNA fragments present in small RNA-seq datasets, namely, Short Uncharacterized RNA Fragment Recognition (SURFR). Briefly, SURFR aligns next generation sequencing (NGS) datasets to a frequently updated database of all human ncRNAs, performs a wavelet analysis to specifically determine the location and expression of ncRNA-derived fragments (ndRNAs), and then conducts an expression analysis to identify significantly differentially expressed ndRNAs [10,11]. We began by utilizing SURFR to determine sdRNA expressions in 489 PCa and 52 normal prostate TCGA patient RNA-seq datasets. This produced a ranked catalogue of significantly differentially expressed sdRNAs in PCa (Appendix A). We elected to focus on sdRNA-A24 and sdRNA-D19b for in vitro characterization as: (1) SdRNA-D19b is expressed (avg. 384 RPM) in 91.6% of 489 TCGA PCa samples versus only 42.3% of normal tissue controls (avg. 162 RPM), and sdRNA-A24 is expressed (avg. 711 RPM) in 97.5% of 489 TCGA PCa samples versus only 30.8% of normal tissue controls (avg. 150 RPM) (Figure 2A). (2) Both sdRNA-A24 and sdRNA-D19b are specifically excised from unique, annotated snoRNA parental loci (Figure 2B). (3) RNA-seq analyses indicate they are both expressed in PC3 cells in agreement with our qRT-PCR analyses (data not shown), where they are also found in association with Ago (Figure 2C,D). In summary, sdRNA-A24 and sdRNA-D19b were ultimately selected for experimental interrogation, as they were the only two sdRNAs found in association with Ago in PC3 cells that were expressed in >90% of TCGA PCa samples but <50% of TCGA normal tissue controls (Appendix A).

### 3.2. sdRNA-D19b and sdRNA-A24 Expressions Directly Affect PC3 Cell Proliferation

We selected the PC3 cell line to interrogate the CRPC sdRNAome and determine whether sdRNAs-D19b and -A24 contribute to the CRPC phenotype. PC3 cells are commonly used as a model of aggressive CRPC, as they do not express the androgen receptor, and their growth is independent of androgen signaling [37]. To manipulate sdRNA expression, we used a custom mimic/inhibitor system detailed and validated in a previous publication from our lab [7]. In brief, RNA sequences identical to sdRNA-D19b and sdRNA-A24 were commercially synthesized and used to simulate sdRNA overexpression through transfecting PC3 cells with these specific sdRNA mimics. Conversely, RNAs complementary to sdRNA-D19b or sdRNA-A24 were similarly synthesized and employed as sdRNA inhibitors through transfecting PC3 cells with these specific sdRNA antagomiRs. We first evaluated the effects of manipulating sdRNAs-D19b and -A24 expressions on PC3 proliferation. Excitingly, the misexpression of either sdRNA-D19b or sdRNA-A24 profoundly impacted PC3 proliferation as compared to control sdRNAs (sdRNA-A61 and sdRNA-93), which are not significantly expressed in TCGA PCa samples, but interestingly, were previously shown to positively contribute to breast cancer cell proliferation [7]. The overexpression of sdRNA-D19b increased PC3 cell proliferation by 24% and 32% at 24 and 72 h, respectively (as compared to cells transfected with scrambled controls). Conversely, sdRNA-D19b inhibition reduced PC3 cell proliferation by 22% and 32% at 24 and 72 h, respectively. Similarly, sdRNA-A24 overexpression enhanced PC3 proliferation by ~25% at both 24 and 72 h, and sdRNA-A24 inhibition decreased proliferation by 14% and 40% at 24 and 72 h, respectively (as compared to cells transfected with scrambled controls). Conversely, PC3 proliferation was not significantly altered following the manipulation of the expressions of two distinct, control sdRNAs expressed in PC3 cells but not differentially expressed in PCa. Collectively, these results indicate functional involvements for both sdRNA-D19b and sdRNA-A24 in PC3 proliferation (Figure 3A).

### 3.3. sdRNA-D19b Overexpression Enhances PC3 Cell Migration

Uncontrolled cell proliferation is a key cellular process during oncogenesis and is recognized as a hallmark of cancer [38]. Another vital hallmark is the acquisition of migratory capabilities, enabling primary tumors to exit their local environment and give rise to metastases. These metastases are primarily responsible for patient mortality [39]. AR-PCa is notoriously metastatic, a characteristic largely responsible for its associated high morbidity. As such, we next assessed whether sdRNAs-D19b and -A24 similarly contribute to PC3 cell migration via the wound-healing assay. In this method, a “scratch” was introduced to bisect confluent cells in a culture dish following sdRNA mimic, inhibitor, or scrambled control transfection (Figure 3B,C) (Appendix A). We found neither sdRNA-D19b, sdRNA-A24 inhibition, nor sdRNA-A24 overexpression significantly altered the PC3 migration as compared to the controls. Notably, we similarly found neither inhibition nor overexpression of a sdRNA significantly overexpressed in TCGA PCa samples (Appendix A) but not expressed in PC3 cells (sdRNA-D42a) significantly altered PC3 migration. In striking contrast, however, we found sdRNA-D19b overexpression markedly increased migration (avg 86.8%) between 6 h and 24 h (Figure 3D).

### 3.4. sdRNA-D19b and sdRNA-A24 Manipulations Alter Drug Sensitivities In Vitro

To assess the potential role of sdRNAs-D19b and -A24 in modulating PCa drug resistance, we examined treatment with three cytotoxic agents, paclitaxel, cisplatin, and dasatinib, to encompass a range of mechanisms of action of drugs typically leveraged to treat CRPC. PC3 cells were treated with one of the chemotherapeutic drugs and either sdRNA mimic, inhibitor, or scrambled control, and then the cells were enumerated every 6 h to assess the impact of sdRNA expression on chemoresistance. Neither overexpression nor inhibition of sdRNA-D19b significantly altered PC3 sensitivity to paclitaxel. In contrast, sdRNA-A24 overexpression improved PC3 resistance to paclitaxel, increasing cell viability between 28.9% and 70.3% at all time points as compared to controls and, although not statistically significant, the sdRNA-A24 inhibition reciprocally sensitized PC3 cells to paclitaxel by 43.2% and 23.9% at 18 and 24 h, respectively (Figure 4A). Conversely, sdRNA-D19b overexpression markedly desensitized PC3 cells to dasatinib treatment, increasing cell viability by over 3-fold at 24 h as compared to controls, whereas neither sdRNA-D19b inhibition nor sdRNA-A24 overexpression nor inhibition produced any discernable effect (Figure 4B). Finally, we found manipulating neither sdRNA-D19b nor sdRNA-A24 levels significantly altered PC3 sensitivity to cisplatin (data not shown). Together, these results clearly support a significant, albeit complex, role for sdRNAs in PC3 drug resistance and strongly imply that sdRNA-D19b and sdRNA-A24 occupy different mechanistic roles in greater drug resistance.

### 3.5. sdRNA-D19b and sdRNA-A24 Target the 3′UTRs of CD44 and CDK12, Respectively

Putative mRNA targets were identified using a strategy previously developed by our group [7] that (1) limits potential targets to those predicted by multiple algorithms and (2) confirms target mRNAs are expressed in PC3 cell RNA-seq datasets. Employing this streamlined methodology readily yielded marked candidates for both sdRNA-D19b and -A24 regulation (Appendix A), and we selected the most notable of these for further validation in vitro. The highest scoring target mRNA identified for sdRNA-D19b (containing two notable 3’UTR complementarities) is a known regulator of PCa proliferation and migration and the cell adhesion glycoprotein CD44 [40] (Figure 5A, top). Similarly, the highest scoring target mRNA identified for sdRNA-A24 (also containing two notable 3’UTR complementarities, one bearing 100% complementarity to sdRNA-A24 nucleotides 2 through 18) is a known tumor suppressor mutated in ~6% of patients with metastatic CRPC, CDK12 [41,42] (Figure 5A, bottom). Importantly, sdRNA-D19b mimic transfection of PC3 cells’ silenced expression from a standard Renilla luciferase reporter containing the principle putative CD44 3′UTR target sites by more than 40%, as compared to the control and sdRNA-A24 mimic transfections. Conversely, the sdRNA-A24 mimic transfection of PC3 cells’ silenced expression from a standard Renilla luciferase reporter containing the principle CDK12 3′UTR target sites by ~70%, as compared to control and sdRNA-D19b mimic transfections (Figure 5B).

## 4. Discussion

PCa was the most prevalent malignancy in American men in 2021 and can broadly be divided into either the androgen-sensitive or castration-resistant phenotype [31,34]. PCa is often readily treatable by therapeutic and surgical interventions to limit the concentration of androgens available to the tumor. Unfortunately, these remissions frequently end with the resurgence of CRPC, a more aggressive and treatment-resistant iteration of the initial cancer [33]. The lack of sustainable treatment options for CRPC largely contributes to the fact that PCa is the second leading cause of cancer death in American men, behind only lung cancer [31].

Numerous miRNAs have now been characterized as master regulators of oncogenes and tumor suppressors [16,17]. With the preponderance of studies implicating miRNAs in virtually all cancer types, aberrant miRNA expression has been rightfully proposed to constitute a hallmark of cancer [18]. Similarly, over the past decade, a growing number of studies have suggested sdRNAs could likewise play significant roles in malignancy [11]. Of note, in 2017, our lab identified sdRNA-93 as a potent inhibitor of breast cancer cell migration [7]. In light of this, we recently explored the potential for sdRNAs to function similarly in other cancer types, leading to the identification and characterization of direct roles for sdRNAs-D19b and -A24 in modulating CRPC.

Importantly, an association between sdRNA misexpression with malignant transformation and metastatic progression in PCa was originally suggested in 2015 by Martens-Uzunova et al. based on their small RNA sequencing of a cohort of 106 matched normal and prostate cancer patient samples. The group identified 319 sdRNAs significantly increased in prostate cancer tissue as compared to normal paired controls. In addition, they found sdRNA-D78 significantly (*p* < 0.0001) upregulated in the cohort that developed metastatic disease, suggesting its potential utility as a prognostic biomarker [11]. However, whereas sdRNAs-D19b and -A24 were identified as being differentially expressed in their analyses, our SURFR analyses do not identify sdRNA-D78 as a likely contributor to PCa. We suggest this is likely due to one (or a combination) of three factors: (1) our analyses focus specifically on CRPC, (2) their alignments allowed for mismatches in read alignments confounding locus assignment, and/or (3) sdRNA-D78 was excluded due to not meeting the minimal expression threshold required by SURFR.

Regardless, while misregulated sdRNAs have been reported in various PCa models, no sdRNAs thought to specifically contribute to the CRPC phenotype have been reported to date [11,21,25]. As such, the work reported herein specifically focused on identifying and characterizing sdRNA misexpressions directly involved with CRPC pathogenesis. We used PC3 cells to assess the impact of sdRNA misexpression, as PC3 cells are widely used as a model of aggressive CRPC, and provide an ideal environment to test our hypothesis that sdRNAs contribute to the CRPC phenotype and their recalcitrance towards therapies [37]. A core characteristic of CRPC is enhanced metastasis, a factor largely responsible for the marked morbidity and high death rate among men in the US [34]. As such, the striking phenotypic consequences associated with manipulating sdRNA-D19b and sdRNA-A24 expressions described in this work (e.g., sdRNA-D19b overexpression results in an ~100% increase in PC3 migration) strongly indicate an important role occupied by sdRNAs in promoting CRPC malignant traits. Of note, however, we observed no effects of manipulating sdRNA-D19b or -A24 levels on cellular invasion (data not shown), and what is more, the current study did not examine the effects of manipulating sdRNA-D19b and -A24 levels in androgen sensitive PCa and therefore does not exclude the possibility that manipulating these sdRNAs may have similar or potentially even distinct phenotypic consequences in castration-sensitive cell models.

In addition to the aggressively metastatic nature of CRPC, this cancer is notoriously difficult to treat. Chemoresistance frustrates treatment regimens for all cancers, but is of particular significance in PCa [28,43]. Prostate tumors are initially responsive to androgen deprivation therapeutics or surgical procedures such as the removal of one or both testes to reduce the androgen concentration accessible for the tumor [44]. Either chemical or surgical castration typically results in disease control and remission lasting 2–3 years. Unfortunately, these remissions commonly end with the resurgence of a more aggressive and treatment-recalcitrant CRPC iteration of the patients’ previous cancer [32]. Notably, our results suggest a marked, hitherto undescribed involvement of sdRNAs in CRPC drug resistance. Excitingly, we find sdRNA-A24 overexpression significantly desensitizes PC3 cells to treatment with the microtubule-stabilizing agent paclitaxel, and the sdRNA-D19b overexpression starkly decreases PC3 sensitivity to dasatinib, a receptor tyrosine kinase (RTK) inhibitor [45]. In addition to implicating sdRNA-D19b and/or sdRNA-A24 as putative drug targets to sensitize PCa to treatment, these results suggest that sdRNAs may be involved with the regulation of core drug resistance components as paclitaxel and dasatinib largely represent mechanistically distinct chemotherapies.

In addition to our phenotypic evaluations, we also elected to explore potential mechanisms of action responsible for the effects associated with sdRNA-D19b and -A24 manipulations. We began by using in silico miRNA target prediction tools to identify potential mRNA targets for sdRNA-D19b and -A24, as (1) our lab [7] and others [8,9,10,11,12] have now described microRNA-like regulations associated with sdRNA expression/mRNA binding, and (2) we find sdRNA-D19b and -A24-enriched in PC3 cell Ago immunoprecipitations. However, whereas additional non-miRNA-like functions of sdRNA-D19b and -A24 cannot be excluded, the ability of these sdRNAs to silence luciferase constructs bearing complementary 3′UTR target sites clearly argues that they function, at least in part, through miRNA-like repressions. That said, accurate target prediction for ncRNAs can prove to be a difficult task, as RNA-target interactions are driven by a number of factors. Common prediction tools typically employ an array of strategies (e.g., miRNA seed sequence complementarity, target site conservation, thermodynamic stability of the predicted interaction, etc.) and as such, each carries distinct advantages and disadvantages [46]. In addition, many target prediction tools routinely predict hundreds of putative targets for individual miRNAs and miRNA-like sdRNAs [47,48,49]. Therefore, we elected to employ a strategy previously developed by our group to prioritize putative targets by (1) limiting potential targets to those predicted by multiple algorithms and (2) confirming target mRNAs are expressed in PC3 cell RNA-seq datasets [7]. Employing this streamlined methodology readily yielded marked candidates for both sdRNA-D19b and -A24 regulation (Appendix A), and reporter assays confirm the ability of sdRNA-D19b and -A24 to repress target sites corresponding to the most notable of these in vitro. Excitingly, the highest scoring target mRNA identified for sdRNA-D19b is a known regulator of PCa proliferation and migration, namely, the cell adhesion glycoprotein CD44 [40] (Figure 5A, top). Similarly, the highest scoring target mRNA identified for sdRNA-A24 is CDK12, a known tumor suppressor mutated in ~6% of patients with metastatic castration-resistant PCa [41,42] (Figure 5A, bottom).

Strikingly, both CD44 and CDK12 are well-defined PCa tumor suppressors that, when downregulated, (1) have clinically-relevant implications and (2) based on our findings, would be expected to be a direct consequence of sdRNA-D19b and -A24 overexpression, respectively. Of note, a loss of CD44 expression is frequently associated with enhanced PCa progression and markedly promotes PCa metastasis [50]. In agreement with this, our work strongly suggests that sdRNA-D19b can directly suppress CD44 expression, and importantly demonstrates that sdRNA-D19b overexpression markedly increases PC3 cell migration in vitro. Also of note, the loss of the sdRNA-A24 target gene CDK12 in CRPC defines a clinically relevant subclass of CRPC that is characteristically hyper-aggressive [41]. CDK12 is a cyclin-dependent kinase that promotes genomic stability through various DNA repair pathways, and a loss of CDK12 expression in PCa enhances genomic mutagenicity, resulting in an aggressive and treatment-resistant phenotype [51]. In this study, we demonstrated that CDK12 is directly regulated by sdRNA-A24, and that sdRNA-A24 overexpression significantly desensitizes PC3 cells to treatment with the microtubule-stabilizing agent, paclitaxel. Interestingly, miR-613 was recently reported to similarly directly modulate paclitaxel resistance via targeting CDK12 in human breast cancer [52].

Therefore, while CD44 and CDK12 likely represent only one of several cellular targets for each sdRNA, this study has redefined the CD44 and CDK12 tumor suppressive axes to include sdRNAs as potent regulators. What is more, we suggest that there are clear clinical ramifications associated with our findings. For example, the CDK12 loss arising from DNA alterations has recently been suggested to represent a powerful new diagnostic for stratifying CRPC patient prognosis [41,42,51]. The work presented here clearly suggests that sdRNA-A24 overexpression can likewise significantly reduce CDK12 expression resulting in a more metastatic cancer phenotype. SdRNA-A24 overexpression functionally mirrors CDK12 deletion, in that the CDK12 protein expression is ablated. As such, any clinical strategy identifying CDK12-deficienct tumors based solely upon genotyping would entirely miss patients with WT CDK12 but overexpressed sdRNA-A24. These patients’ cancers would be expected to manifest the same phenotypic properties and, of critical importance, sensitivities or resistances to therapeutic interventions. As such, the development of effective CDK12-based diagnostics will likely require protein-level evaluation and/or CDK12 genotyping coupled with small RNA sequencing.

Finally of note, in 2019, McMahon et al. demonstrated that specific subsets of snoRNAs are differentially regulated during the earliest cellular response to oncogenic RAS^G12V^ in mice, and that a loss of SNORA24 cooperates with RAS^G12V^ to promote the development of liver cancer closely resembling human steatohepatitic hepatocellular carcinoma (HCC). Notably, they found that human HCCs characterized by low SNORA24 expression are significantly associated with poor patient survival [53]. Although seemingly contradictory, we suggest the (1) reported association between a loss of functional, full length SNORA24 and HCC development, and (2) the positive contribution of increased sdRNA-A24 excision/expression in CRPC that we report here may actually well agree. Although clearly an oversimplification, in the event of a finite, fixed amount of the SNORA24 precursor, increasing sdRNA24 excision/expression would directly result in a loss of functional full length SNORA24. Regardless, further study is required to determine if the overexpression of these sdRNAs is sufficient to promote CRPC progression, or if instead both overexpression of an sdRNA and concurrent loss of its corresponding full length snoRNA are required.

In summary, with tools such as SURFR [14,15] having only recently made the intensive interrogation of sdRNAomes widely available, we suggest that the identification of relevant sdRNA contributions to malignancy will accelerate in the near future and lead to the development of novel therapies and diagnostics based on sdRNAs. It is important to note, however, that while we find sdRNA-D19b and -A24 significantly more highly expressed in TCGA PCa patient samples than in normal tissue controls, and that manipulating the expressions of these sdRNAs in PC3 cells outlines a biologically coherent mechanism by which sdRNAs downregulate tumor suppressors in AR-PCa to enhance proliferative and metastatic capabilities and to encourage chemotherapeutic resistance, direct validation, and characterization of sdRNA-D19b and/or -A24 misexpressions (in addition to larger patient sample cohorts) will clearly be required to establish the utility of one or both of these sdRNAs as viable biomarkers. In short, considerably more extensive groundwork must be laid before these (or any) sdRNAs can be fashioned as tractable drug targets for cancer therapy or as diagnostic/prognostic markers similar to cutting-edge miRNA translational applications [53,54,55]. That said, we do suggest the work presented here does begin to expand the CRPC regulatory landscape to include sdRNAs as potential new therapeutic targets and/or prognostic indicators through identifying sdRNA-D19b and sdRNA-A24 as likely contributors to CRPC, an aggressive molecular subtype of PCa for which there are currently only limited options for therapy.

## Figures and Tables

**Figure 1 cells-11-01302-f001:**
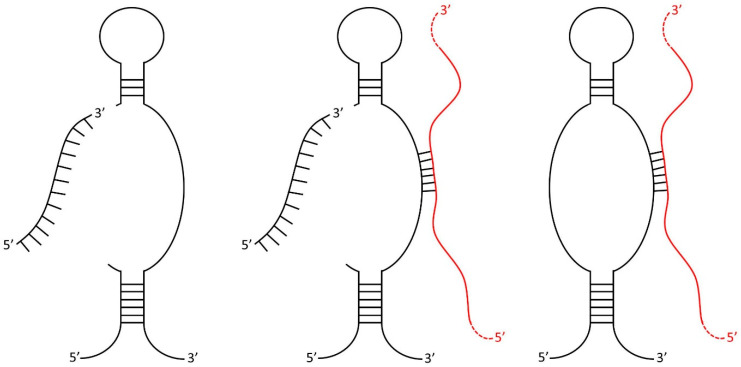
SdRNAs are specifically processed from annotated snoRNA loci. Transcripts arising from various annotated snoRNA loci have now been definitively shown to participate in at least two distinct noncoding RNA regulatory pathways. Individual loci can produce snoRNAs functioning exclusively as traditional posttranscriptional RNA editors directing 2′-O-methylation or pseudouridylation of transcripts (**right**), or exclusively as functional miRNA precursors (**left**). Some loci have now been confirmed to produce transcripts, at times engaging in both types of noncoding RNA regulation (**center**). MiRNA-like excision products are illustrated in black (**left** and **center**) as excision products of primary transcript. Complementary RNA editing targets are shown in red (**right** and **center**) with red dashes indicating the larger transcript excluded for the purpose of clarity. Adapted from Patterson et al. [7].

**Figure 2 cells-11-01302-f002:**
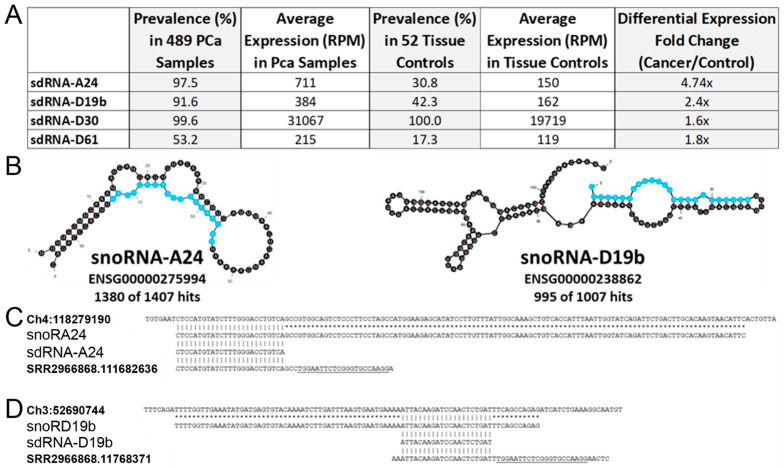
SdRNAs-D19b and -A24. (**A**) SdRNA-A24 and sdRNA-D19b are significantly overexpressed sdRNAs in TCGA prostate cancer patient datasets. The SURFR algorithm [14,15] was used to identify sdRNAs abundantly expressed in prostate cancer patient tumors versus normal prostate. (**B**) The most thermodynamically stable secondary structures of putative sdRNA producing snoRNAs with sdRNA sequences highlighted in blue as calculated by Mfold [36]. Common name and Ensembl gene ID for putatively processed snoRNAs are listed below corresponding structures. “Hits” refer to the number of times fragments of putative sdRNA producing snoRNAs perfectly aligned to small RNA-seq reads (PRAD ID: f45a166f-d67b-5de1-8cbd-b5782659457a) from the TCGA prostate cancer dataset. Numbers preceding total numbers of hits correspond to the number of times positions highlighted in blue (putative sdRNAs) perfectly aligned to small RNA-seq reads (e.g., 1380 of 1407 small RNA reads aligning to snoRNA-A24 corresponded to the sequence highlighted in blue). (**C**) Alignment between the human genome (GRCh38:chr4:118279190-118279320:1) (top), SNORA24 (ENSG00000275994) (upper middle), sdRNA-A24 (SURFR call) (lower middle), and next generation small RNA sequence read (bottom) obtained by Illumina sequencing of PC3 cell Ago immunoprecipitations (SRR2966868) is shown. The underlined sequence corresponds to the Illumina TruSeq Small RNA adapter RA3. All sequences are in the 5′ to 3′ direction. An asterisk indicates base identity between the snoRNA and genome. Vertical lines indicate identity across all three sequences. (**D**) Alignment (as in (**C**)) between the human genome (GRCh38:chr3:52690744-52690827:1) (top), SNORD19b (ENSG00000238862) (upper middle), sdRNA-D19b (SURFR call) (lower middle), and next generation small RNA sequence read (bottom) obtained by Illumina sequencing of PC3 cell Ago immunoprecipitations.

**Figure 3 cells-11-01302-f003:**
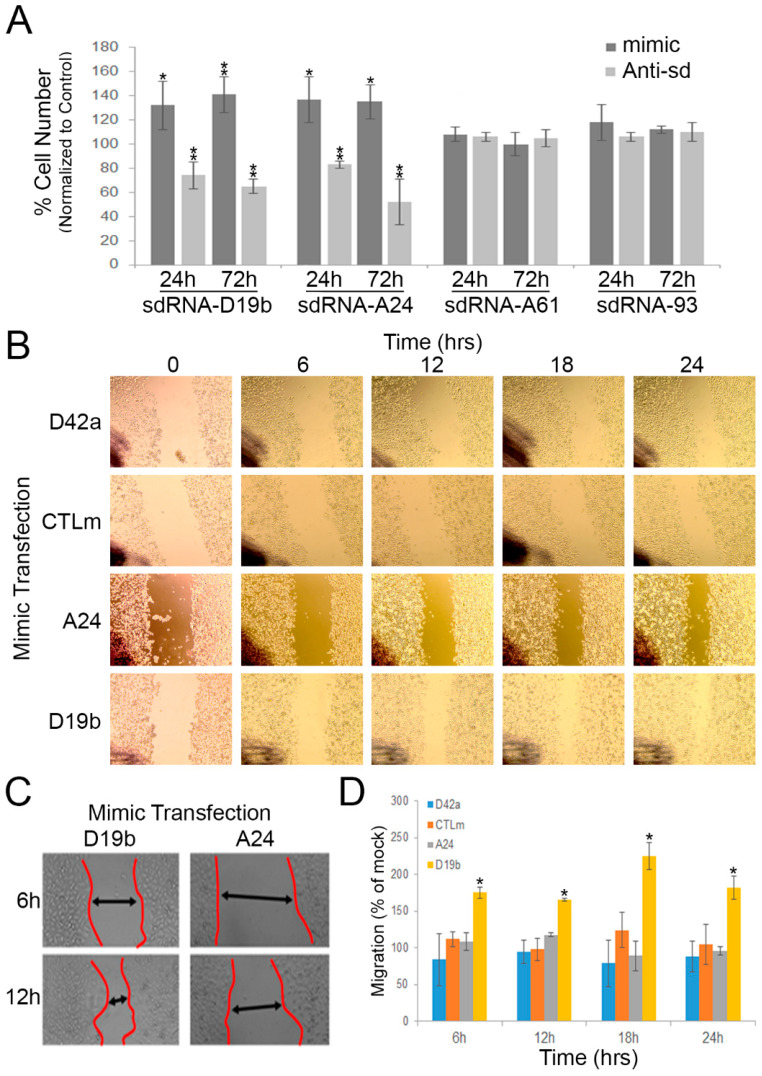
SdRNA-D19b and -A24 levels significantly impact PC3 cell proliferation and migration. (**A**) PC3 cells were transfected with indicated sdRNA mimic or antagomiR (Anti-sd). Cell counts were performed at 24 and 72 h then normalized to scrambled control transfections (*n* = 8). * indicates *p* ≤ 0.05; ** indicates *p* ≤ 0.01; *p*-values by unpaired two-tailed *t*-test. (**B**,**C**) Representative migration (wound-healing) assays for PC3 cells transfected with the indicated sdRNA mimic. Wound border closure is indicated by black arrows. (**D**) PC3 migration assays quantified. Images were captured at the indicated times (*X*-axis) and wound healing quantified using ImageJ as % migration normalized to scrambled control (*n* ≥ 3). * indicates *p* ≤ 0.05; *p*-values by unpaired two-tailed *t*-test. D42a, sdRNA-D42a mimic; CTLm, scrambled mimic; A24, sdRNA-A24 mimic; D19b, sdRNA-D19b mimic.

**Figure 4 cells-11-01302-f004:**
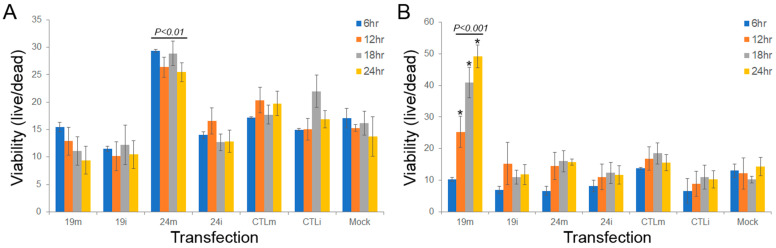
SdRNA overexpression protects PC3 cells from chemotherapeutic agents. Cells were cultured in 24-well plates and transfected at 70% confluency with mimics or inhibitors. Following transfection, cells were treated with (**A**) paclitaxel (5 nM) or (**B**) dasatinib (50 nM). Cell death was quantified every 6 h for 24 h total using ImageJ and methylene blue dead cell staining. 19 m, sdRNA-D19b mimic; 19i, sdRNA-D19b inhibitor; 24 m, sdRNA-A24 mimic; 24i, sdRNA-A24 inhibitor; CTLm, scrambled mimic; CTLi, scrambled inhibitor; Mock, vehicle-treated control. (*n* ≥ 3). * indicates *p* < 0.001; *p*-values by unpaired two-tailed *t*-test as compared to Mock.

**Figure 5 cells-11-01302-f005:**
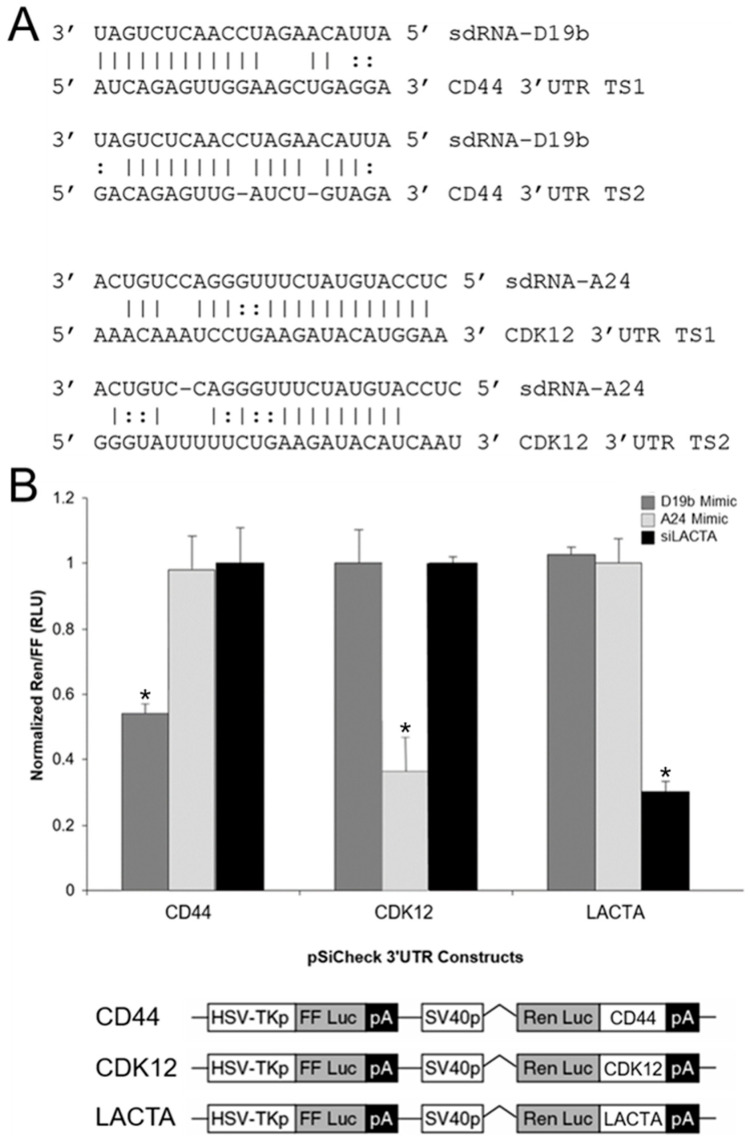
SdRNA-D19b and sdRNA-A24 mRNA targets. (**A**) Alignments between putative 3′UTR target sites with sdRNAs-D19b (top) and -A24 (bottom). Vertical lines indicate Watson-Crick basepair. Dotted lines indicate G:U basepair. TS1, target site 1. TS2, target site 2. (**B**) SdRNAs-D19b and -A24 specifically repress luciferase expression from mRNAs containing CD44 and CDK12 target sites in their 3′UTRs. SdRNA mimics and luciferase reporters with target sequences (bottom) and/or controls (LACTA refers to beta galactosidase control sequence) were constructed and cotransfected, as previously described [7]. * indicates *p* < 0.01; *p*-values by unpaired two-tailed *t*-test as compared to LACTA excepting LACTA compared to CD44.

## Data Availability

All next-generation small RNA deep-sequencing libraries utilized are publicly available and were obtained from NCBI SRA. SRR Files analyzed: SRR2966868, SRR2966869, SRR3502951, SRR3502954, SRR3502977, SRR12617203, and SRR12617204. The TCGA datasets utilized during the current study are all available in the NCI’s Genomic Data Commons (GDC, https://portal.gdc.cancer.gov/) (accessed on 14 January 2019). These include the raw data used to generate all figures and statistical analysis. While the majority of the data used in the paper is open-access, access to TCGA protected MAFs from the GDC requires dbGaP approval (phs000178, https://www.ncbi.nlm.nih.gov/projects/gap/cgi-bin/study.cgi?study_id=phs000178.v11.p8) (accessed on 14 January 2019). All other relevant data (e.g., alignment files) are available from the authors upon request.

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
