# Peer review of "MicroRNA-like snoRNA-Derived RNAs (sdRNAs) Promote Castration-Resistant Prostate Cancer"

_cells, 2022, doi:10.3390/cells11081302_

Round 1
Reviewer 1 Report
In this manuscript, the authors investigated snoRNA fragments using TCGA prostate cancer (PCa) data and identified two potential biomarkers and therapeutic targets for AR-PCa by analyzing differential sdRNAs, but there are many questions in this manuscript
Major comments:
- The pictures in the article are blurry and unrecognizable
- Figure 3B in the article lacks the normal group as a control, and the supplementary file does not see the difference between the normal group and sdRNA-d19b and sdRNA-a24.
- How to screen out the two sdRNAs in the article? The FC of the two sdRNAs selected in the article is not the largest, or even lower.
- Histograms in the article lack significance scores, such as Figure3C and Figure4
- There is less content in the article, experiments and computer analysis are not enough, and the discussion part accounts for the majority.
- To screen mRNAs complementary to sdRNA (Figure5), the specific alignment scores including other candidate mRNAs need to be listed.
- Have other PCa cell lines been validated? Is it possible that these 2 sdRNAs affect both PCa cells at all stages, not just CRPC?
Author Response
We would 1st like to thank the reviewer for their thoughtful critique of our work. We have addressed each of their concerns to the best of our ability as follows:
Major comments:
1. The pictures in the article are blurry and unrecognizable
We apologize for any difficulty this created during review and have replaced the original images with much higher resolution revisions.
2. Figure 3B in the article lacks the normal group as a control, and the supplementary file does not see the difference between the normal group and sdRNA-d19b and sdRNA-a24.
We have greatly expanded Figure 3 to include 2 new controls and 3 additional time points. What’s more we have added images corresponding to all 5 time points for 4 normal mock wells as Supplementary File 3.i.
3. How to screen out the two sdRNAs in the article? The FC of the two sdRNAs selected in the article is not the largest, or even lower.
This has been directly addressed by expanding Supplementary File 2A to include % Cancer File Expression, % Control File Expression, and percentage expression changes and revising the text to include the following: “We elected to focus on sdRNA-A24 and sdRNA-D19b for in vitro characterization as: (1) SdRNA-D19b is expressed (avg. 384 RPM) in 91.6% of 489 TCGA PCa samples versus only 42.3% of normal tissue controls (avg. 162 RPM), and sdRNA-A24 is expressed (avg. 711 RPM) in 97.5% of 489 TCGA PCa samples versus only 30.8% of normal tissue controls (avg. 150 RPM) (Figure 2A). (2) Both sdRNA-A24 and sdRNA-D19b are specifically excised from unique, annotated snoRNA parental loci (Figure 2B). And (3) RNA-seq analyses indicate they are both expressed in PC3 cells in agreement with our qRT-PCR analyses (data not shown) where they are also found in association with Ago (Figure 2C,D). In summary, sdRNA-A24 and sdRNA-D19b were ultimately selected for experimental interrogation as they were the only two sdRNAs found in association with Ago in PC3 cells that were expressed in >90% of TCGA PCa samples but <50% of TCGA normal tissue controls (Supplementary File 2A).”
4. Histograms in the article lack significance scores, such as Figure3C and Figure4
P-values have been calculated and are now included for Figures 3A, 3D(formerly 3C), 4A, 4B, and 5B.
5. There is less content in the article, experiments and computer analysis are not enough, and the discussion part accounts for the majority.
As indicated above, additional scratch assay images have been added to Figure 3 and Supplementary File 3. What’s more, additional controls have been added to Figures 3 and 4, and a table detailing all the sdRNA expressions in 499 TCGA PCa and 52 control patient samples is now included as Supplementary File 2B. Descriptions of each of these additions are now included in the revised Results section.
6. To screen mRNAs complementary to sdRNA (Figure5), the specific alignment scores including other candidate mRNAs need to be listed.
Best alignment lengths, percent complementarities, and the presence of multiple target sites has been added to “Supplementary File 4. sdRNA -D19b and -A24 complementary mRNAs.”
7. Have other PCa cell lines been validated? Is it possible that these 2 sdRNAs affect both PCa cells at all stages, not just CRPC?
This has been directly addressed in the revised discussion as follows: “As such, the striking phenotypic consequences associated with manipulating sdRNA-D19b and sdRNA-A24 expressions described in this work (e.g., sdRNA-D19b overexpression results in an ~100% increase in PC3 migration) strongly indicate an important role occupied by sdRNAs in promoting CRPC malignant traits. Of note, however, we observed no effects of manipulating sdRNA -D19b or –A24 levels on cellular invasion (data not shown), and what’s more, the current study did not examine the effects of manipulating sdRNA -D19b and –A24 levels in androgen sensitive PCa and therefore does not exclude the possibility that manipulating these sdRNAs may have similar or potentially even distinct phenotypic consequences in castration sensitive cell models.”
Reviewer 2 Report
The authors present a highly original study demonstrating the presence of differentially expressed snoRNA derived fragments from prostate tumor tissues and their role in the growth and proliferation of prostate cancer cells by leveraging their in-house built bioinformatics tool SURFR. The authors establish a coherent framework for how these fragments may influence cell growth via a miRNA like action on downstream genes such as CD44 and CDK12. Overall, the study is rigorously performed, novel, and well written. Some minor suggestions that would improve the presentation of the work and help to clarify some points:
- The computationally predicted targets of sdRNAs was explained as being very large based upon a miRNA seed algorithm approach and reuse of miRNA tools. The authors then established a clever method to greatly reduce the number of candidates. This is reasonable, however, a theoretical rationale for why sdRNAs should be “miRNA like” was not clarified in detail. The obvious explanation would be size, but many other ncRNA fragments are also in this range. While binding to AGO proteins was demonstrated (Fig2C,D), this cannot rule out non-miRNA like mechanisms. Authors should better justify the theoretical approach to target prediction.
- The rationale for choosing to focus on sdRNA-D19b and -A24 is not readily clear as many of the other 38 identified sdRNAs have higher ratios or expression levels (Supplementary File 2).
- Figure 1 seems cryptic especially in what is meant by “editor”. RNA editing, folding, epigenetic editing ? A clearer and more detailed legend should be provided so readers can grasp the concept more easily.
- Figures 2,3,4 are low resolution, they should be upgraded.
- Do CD44 or CDK12 expression levels correlate with survival in PCa from TCGA data sets ? Related to this, the mechanistic studies were performed on cell cultures whereas the original data came from patient tumour tissues collected in the TCGA database. Authors should comment on limits to extrapolations in the discussion. For example, in the abstract, if these sdRNAs were to be useful biomarkers, then some validation should be done on human tissues. It is not suggested to do it in this manuscript, but this notion should be acknowledged in the discussion if the abstract statement is to be supported.
- “data availability” should be capitalized.
- “That said” line 249, line 261, line 276, line 289, line 305, line 334, line 360, line 369 is overused.
Author Response
We would 1st like to thank the reviewer for their thoughtful critique and generally positive review (“Overall, the study is rigorously performed, novel, and well written.”) of our work. We have addressed each of their concerns to the best of our ability as follows:
Some minor suggestions that would improve the presentation of the work and help to clarify some points:
- The computationally predicted targets of sdRNAs was explained as being very large based upon a miRNA seed algorithm approach and reuse of miRNA tools. The authors then established a clever method to greatly reduce the number of candidates. This is reasonable, however, a theoretical rationale for why sdRNAs should be “miRNA like” was not clarified in detail. The obvious explanation would be size, but many other ncRNA fragments are also in this range. While binding to AGO proteins was demonstrated (Fig2C,D), this cannot rule out non-miRNA like mechanisms. Authors should better justify the theoretical approach to target prediction.
This has been directly addressed in the revised discussion as follows: “In addition to our phenotypic evaluations, we also elected to explore potential mechanisms of action responsible for the effects associated with sdRNA-D19b and -A24 manipulations. We began by using in silico miRNA target prediction tools to identify potential mRNA targets for sdRNA-D19b and -A24 as (1) our lab7 and others8–13 have now described microRNA-like regulations associated with sdRNA expression/mRNA binding and (2) we find sdRNA-D19b and –A24 enriched in PC3 cell Ago immunoprecipitations. However, whereas additional non-miRNA like functions of sdRNA-D19b and -A24 can not be excluded, the ability of these sdRNAs to silence luciferase constructs bearing complementary 3’UTR target sites clearly argues that they function, at least in part, through miRNA-like repressions.”
- The rationale for choosing to focus on sdRNA-D19b and -A24 is not readily clear as many of the other 38 identified sdRNAs have higher ratios or expression levels (Supplementary File 2).
This has been directly addressed by expanding Supplementary File 2A to include % Cancer File Expression, % Control File Expression, and percentage expression changes and revising the text to include the following: “We elected to focus on sdRNA-A24 and sdRNA-D19b for in vitro characterization as: (1) SdRNA-D19b is expressed (avg. 384 RPM) in 91.6% of 489 TCGA PCa samples versus only 42.3% of normal tissue controls (avg. 162 RPM), and sdRNA-A24 is expressed (avg. 711 RPM) in 97.5% of 489 TCGA PCa samples versus only 30.8% of normal tissue controls (avg. 150 RPM) (Figure 2A). (2) Both sdRNA-A24 and sdRNA-D19b are specifically excised from unique, annotated snoRNA parental loci (Figure 2B). And (3) RNA-seq analyses indicate they are both expressed in PC3 cells in agreement with our qRT-PCR analyses (data not shown) where they are also found in association with Ago (Figure 2C,D). In summary, sdRNA-A24 and sdRNA-D19b were ultimately selected for experimental interrogation as they were the only two sdRNAs found in association with Ago in PC3 cells that were expressed in >90% of TCGA PCa samples but <50% of TCGA normal tissue controls (Supplementary File 2A).”
- Figure 1 seems cryptic especially in what is meant by “editor”. RNA editing, folding, epigenetic editing ? A clearer and more detailed legend should be provided so readers can grasp the concept more easily.
We appreciate the reviewer’s suggestions to better convey these points to the reader. In addition to revising Figure 1, the figure legend has been considerably revised and now reads: “Figure 1. SdRNAs are specifically processed from annotated snoRNA loci. SdRNAs are specifically processed from annotated snoRNA loci. Transcripts arising from various annotated snoRNA loci have now been definitively shown to participate in at least two distinct noncoding RNA regulatory pathways. Individual loci can produce snoRNAs functioning exclusively as traditional posttranscriptional RNA editors directing 2'-O-methylation or pseudouridylation of transcripts (right), or exclusively as functional miRNA precursors (left). Some loci have now been confirmed to produce transcripts at times engaging in both types of noncoding RNA regulation (center). MiRNA-like excision products are illustrated in black (left and center) as excision products of primary transcript. Complementary RNA editing targets are shown in red (right and center) with red dashes indicating the larger transcript excluded for the purpose of clarity. Adapted from Patterson et. al7”
- Figures 2,3,4 are low resolution, they should be upgraded.
We apologize for any difficulty this created during review and have replaced the original images with much higher resolution revisions.
- Do CD44 or CDK12 expression levels correlate with survival in PCa from TCGA data sets ? Related to this, the mechanistic studies were performed on cell cultures whereas the original data came from patient tumour tissues collected in the TCGA database. Authors should comment on limits to extrapolations in the discussion. For example, in the abstract, if these sdRNAs were to be useful biomarkers, then some validation should be done on human tissues. It is not suggested to do it in this manuscript, but this notion should be acknowledged in the discussion if the abstract statement is to be supported.
Unfortunately, the clinical data associated with the TCGA PCa data is largely incomplete in terms of tracking patient outcome. That said, we agree with the reviewer that the limits to extrapolations should be included. As such, this is now included in the revised discussion as follows: “It is important to note, however, that while we find sdRNA-D19b and -A24 significantly more highly expressed in TCGA PCa patient samples than in normal tissue controls, and that manipulating the expressions of these sdRNAs in PC3 cells outlines a biologically coherent mechanism by which sdRNAs downregulate tumor suppressors in AR- PCa to enhance proliferative and metastatic capabilities and to encourage chemotherapeutic resistance, direct validation and characterization of sdRNA-D19b and/or -A24 misexpressions in additional patient sample cohorts will clearly be required to establish the utility of one or both of these sdRNAs as viable biomarkers. In short, considerably more extensive groundwork must be laid before these (or any) sdRNAs can be fashioned as tractable drug targets for cancer therapy or as diagnostic / prognostic markers similar to cutting-edge miRNA translational applications58,59.”
- “data availability” should be capitalized. AND
- “That said” line 249, line 261, line 276, line 289, line 305, line 334, line 360, line 369 is overused.
These points have been corrected in the revised text.
Reviewer 3 Report
The manuscript by Coley and colleagues identifies two snoRNA-derived small RNAs as altered in prostate cancer patients and suggests that they may have a functional role in prostate cancer development. The authors found that the modulation of the expression levels of these two miRNAs, by exogenous expression of specific mimics or antisenses, alters cell proliferation, migration capability and drug resistance of PC3 prostate cancer cell line.
This work is certainly of interest in the view of better characterizing the molecular determinants that fuel prostate cancer progression. However, I believe that the expermiments performed have some flaws that need to be fixed to properly interpret the results. Specifically, these are my major concerns:
Figure 3A: Please indicate if the observed differences are statistically significant. Two additional sdRNAs have been included without having been mentioned in the text. Please justify the rationale of their inclusion (that I suppose should function as negative controls) and the reason beyond their choice among the plethora of possible sdRNAs.
Figure 3B: Please include in the panel representative images of scrambled controls and of all the conditions tested to allow a the proper interpretation of the experiment.
Figure 4: Vehicle-treated controls are missing for each condition assayed. Therefore, it is impossible to discriminate the specific cytotoxicity of the drug from any other nonspecific toxic effects (in example due to transfection). Please provide vehicle-treated controls and use them as references to normalize each collected time point.
Figures are barely readable. In general you should increase the resolution of all the figures as well as the font sizes.
In addition, I have few minor points that should be taken into account:
Figure 1: the size of the red target RNA is too short and misleading (as snoRNAs should be shorter than their target RNAs). It might be useful to represent the extremities with dashed lines.
Please put the complete table as in Figure 2A in supplementary information section.
Line 145: it is not immediately clear the meaning of simulating the overexpression. For clarity please mention in the main text that PC3 cells were transfected with the synthesized sense (or antisense) sdRNAs under investigation (as indicated in the figure legend).
Figure 5B: Change the order of the legend items to match the order of the columns in the plot.
For these reasons I suggest to evaluate a modified version of the manuscript after a round of major revisions.
Author Response
We would 1st like to thank the reviewer for their thoughtful critique of our work. We have addressed each of their concerns to the best of our ability as follows:
Specifically, these are my major concerns:
-Figure 3A: Please indicate if the observed differences are statistically significant. Two additional sdRNAs have been included without having been mentioned in the text. Please justify the rationale of their inclusion (that I suppose should function as negative controls) and the reason beyond their choice among the plethora of possible sdRNAs.
P-values have been calculated and are now included for Figures 3A, 3D(formerly 3C), 4A, 4B, and 5B. In addition, the inclusion of the 2 additional sdRNAs in Figure 3A has been directly addressed in the text as follows: “Excitingly, misexpression of either sdRNA-D19b or sdRNA-A24 profoundly impacted PC3 proliferation as compared to control sdRNAs (sdRNA-A61 and sdRNA-93) which are not significantly expressed in TCGA PCa samples, but interestingly, were previously shown to positively contribute to breast cancer cell proliferation7. Overexpression of sdRNA-D19b increased PC3 cell proliferation by 24% and 32% at 24 and 72 h respectively (as compared to cells transfected with scrambled controls).”
-Figure 3B: Please include in the panel representative images of scrambled controls and of all the conditions tested to allow proper interpretation of the experiment.
We have greatly expanded Figure 3 to include 2 new controls (including scrambled) and 3 additional time points. What’s more we have added images corresponding to all 5 time points for 4 normal mock wells as Supplementary File 3.i.
-Figure 4: Vehicle-treated controls are missing for each condition assayed. Therefore, it is impossible to discriminate the specific cytotoxicity of the drug from any other nonspecific toxic effects (in example due to transfection). Please provide vehicle-treated controls and use them as references to normalize each collected time point.
As suggested, vehicle-treated mock controls have been added to Figure 4 and utilized for determining statistically significant differences.
-Figures are barely readable. In general you should increase the resolution of all the figures as well as the font sizes.
We apologize for any difficulty this created during review and have replaced the original images with much higher resolution revisions.
-In addition, I have few minor points that should be taken into account:
-Figure 1: the size of the red target RNA is too short and misleading (as snoRNAs should be shorter than their target RNAs). It might be useful to represent the extremities with dashed lines.
We appreciate the reviewer’s suggestion to improve this figure for the reader. In addition to revising Figure 1 as suggested, the figure legend has been considerably improved and now reads: “Figure 1. SdRNAs are specifically processed from annotated snoRNA loci. SdRNAs are specifically processed from annotated snoRNA loci. Transcripts arising from various annotated snoRNA loci have now been definitively shown to participate in at least two distinct noncoding RNA regulatory pathways. Individual loci can produce snoRNAs functioning exclusively as traditional posttranscriptional RNA editors directing 2'-O-methylation or pseudouridylation of transcripts (right), or exclusively as functional miRNA precursors (left). Some loci have now been confirmed to produce transcripts at times engaging in both types of noncoding RNA regulation (center). MiRNA-like excision products are illustrated in black (left and center) as excision products of primary transcript. Complementary RNA editing targets are shown in red (right and center) with red dashes indicating the larger transcript excluded for the purpose of clarity. Adapted from Patterson et. al7”
-Please put the complete table as in Figure 2A in supplementary information section.
A table detailing all the sdRNA expressions in 489 TCGA PCa and 52 control patient samples is now included as Supplementary File 2B.
-Line 145: it is not immediately clear the meaning of simulating the overexpression. For clarity please mention in the main text that PC3 cells were transfected with the synthesized sense (or antisense) sdRNAs under investigation (as indicated in the figure legend).
This has been directly addressed in the revised text as follows: “In brief, RNA sequences identical to sdRNA-D19b and sdRNA-A24 were commercially synthesized and used to simulate sdRNA overexpression through transfecting PC3 cells with these specific sdRNA mimics. Conversely, RNAs complementary to sdRNA-D19b or sdRNA-A24 were similarly synthesized and employed as sdRNA inhibitors through transfecting PC3 cells with these specific sdRNA antagomiRs.”
-Figure 5B: Change the order of the legend items to match the order of the columns in the plot.
We appreciate the reviewer catching this point and have corrected it.
Reviewer 4 Report
In this publication Coley et al. describe the roles of microRNA-like snoRNA-derived RNAs (sdRNAs) in castration resistant prostate cancer.
General comment: The topic is relevant and interesting and the article is well written.
Specific points:
Figure 2c and d: The text can't be read and the font size should be increased.
Figure 3 is blurred and the text is hard to read.
Figure 3b: the controls should be shown.
Figure 4 is also blurred and the font size should be increased.
GeneCopoeia is misspelled.
Author Response
We would 1st like to thank the reviewer for their thoughtful critique and generally positive review (“General comment: The topic is relevant and interesting and the article is well written”) of our work. We have addressed each of their concerns to the best of our ability as follows:
Specific points:
Figure 3b: the controls should be shown.
We have greatly expanded Figure 3 to include 2 new controls and 3 additional time points. What’s more we have added images corresponding to all 5 time points for 4 normal mock wells as Supplementary File 3.i.
Figure 2c and d: The text can't be read and the font size should be increased.
Figure 3 is blurred and the text is hard to read.
Figure 4 is also blurred and the font size should be increased.
We apologize for any difficulty this created during review and have replaced the original images with much higher resolution revisions with increased font sizes.
GeneCopoeia is misspelled.
We appreciate the reviewer catching this point and have corrected it.
Round 2
Reviewer 1 Report
If the authors try to show that the small RNA derived from snoRNA has a microRNA-like feature, then the mechanism of its occurrence and functional mode should be demonstrated. For example, verification of the amount of this small RNA after knockdown of Dicer, and demonstration of the ability of this small RNA to bind to AGO proteins and target genes using CLiP-Seq and other experimental methods. Currently, I have not seen any evidence in this regard. In addition, We could not find the expression of two candidate sdRNAs in cell lines from the manuscript, only from TCGA tissues, please provide information on the verification of these sdRNAs in cell lines.
Author Response
- If the authors try to show that the small RNA derived from snoRNA has a microRNA-like feature, then the mechanism of its occurrence and functional mode should be demonstrated. For example, verification of the amount of this small RNA after knockdown of Dicer, and demonstration of the ability of this small RNA to bind to AGO proteins and target genes using CLiP-Seq and other experimental methods. Currently, I have not seen any evidence in this regard.
Respectfully, we suggest this is not a study focused on delineating the mechanisms of action/expression of sdRNAs versus microRNAs and that confirming their association with Ago as well as showing that they can silence the expressions of mRNAs bearing complementary target sites is sufficient to conclude they likely function as microRNAs. Most importantly, regardless of the pathways leading to their expression, our TCGA data analyses clearly support a role for their misexpression in PCa pathology.
That said, we do agree with the reviewer that the proposed role of these sdRNAs as microRNAs could benefit from better describing previous findings supporting this and have therefore added the following to the introduction:
“In 2008, however, widespread, specific processing of snoRNAs into 16-36 nucleotide (nt) fragments largely indistinguishable from microRNAs (miRNAs) was first reported1. SdRNA association with Ago and significantly reduced sdRNA expression in the absence of Dicer 1 was originally demonstrated by Ender et al. in 20087 then again by Taft et al. in 20092. Since these initial publications, numerous studies have similarly confirmed sdRNA Ago and/or Dicer-dependent mRNA silencing with our lab8 and others2,7,9–15 independently reporting findings indicating that sdRNAs are processed analogously to and function as miRNAs (Figure 1).”
- In addition, We could not find the expression of two candidate sdRNAs in cell lines from the manuscript, only from TCGA tissues, please provide information on the verification of these sdRNAs in cell lines.
We have endeavored to better convey this point in the revised manuscript as: “RNA-seq analyses directly confirm they are both expressed in PC3 cells… where they are also found in association with Ago (Figure 2C,D)“. That said, the manuscript directly highlights this point in Figures 2C and 2D:
Figure 2C shows a direct alignment with sdRNA-A24 and an actual read from the SRA “Argonaute protein pull down after UV cross linking following by sequencing of Argonaute bound RNAs in PC3 cells”
SRR2966868.111682636
Likewise, Figure 2D shows a direct alignment with sdRNA-D19b and an actual read from the SRA “Argonaute protein pull down after UV cross linking following by sequencing of Argonaute bound RNAs in PC3 cells”
SRR2966868.11768371
As such, although more sophisticated methods are available, we suggest that the data provided should allow virtually any reader in the field to readily confirm the existence and abundance of these specific sdRNAs in PC3 cell Ago pulldowns by blasting the sdRNAs sequences against SRR2966868 using the standard Web-based NCBI blast interface (https://blast.ncbi.nlm.nih.gov/) and selecting sequence read archive under database and entering the SRR number.
Reviewer 3 Report
I thank the authors for having provided answers to most of the points I raised.
I still have few concerns related to the statistics and the drug treatment experiment performed. Regarding the first, it is not clear from the plots the identity of the two groups that were compared to calculate each p-value, therefore it is hard for the reader to understand whether the data observed are statistically significant. About the latter, a vehicle-treated control is missing for each transfection condition (I mean cells that have been transfected together with the ones that undergo drug treatment but that are treated with the vehicle alone). These controls should be used to normalise the results coming from each treatment performed on the same transfectant cells. The lack of this information does not allow a proper interpretation of the experiment.
For these reason I believe that the authors should address these points before considering this manuscript as suitable for publication.
Author Response
I thank the authors for having provided answers to most of the points I raised. I still have few concerns related to the statistics and the drug treatment experiment performed. Regarding the first, it is not clear from the plots the identity of the two groups that were compared to calculate each p-value, therefore it is hard for the reader to understand whether the data observed are statistically significant. About the latter, a vehicle-treated control is missing for each transfection condition (I mean cells that have been transfected together with the ones that undergo drug treatment but that are treated with the vehicle alone). These controls should be used to normalise the results coming from each treatment performed on the same transfectant cells. The lack of this information does not allow a proper interpretation of the experiment.
We appreciate the reviewer’s comments about the importance of these points. Although we had included this data in our 1st revision, we have now endeavored to better clarify these in the revised figure legends as follows:
Figure 3. SdRNA -D19b and -A24 levels significantly impact PC3 cell proliferation and migration. (A) PC3 cells were transfected with indicated sdRNA mimic or antagomiR (Anti-sd). Cell counts were performed at 24 and 72 h then normalized to scrambled control transfections (n=8). * indicates P≤0.05; ** indicates P≤0.01; P-values by unpaired two-tailed t-test as compared to mock vehicle-treated control. (B,C) Representative migration (wound-healing) assays for PC3 cells transfected with the indicated sdRNA mimic. Wound border closure is indicated by black arrows. (D) PC3 migration assays quantified. Images were captured at the indicated times (X-axis) and wound healing quantified using ImageJ as % migration normalized to scrambled control (n≥3). * indicates P≤0.05; P-values by unpaired two-tailed t-test as compared to mock vehicle-treated control (Supplementary File 3.I.). D42a, sdRNA-D42a mimic; CTLm, scrambled mimic; A24, sdRNA-A24 mimic; D19b, sdRNA-D19b mimic.
Figure 4. SdRNA overexpression protects PC3 cells from chemotherapeutic agents. Cells were cultured in 24 well plates and transfected at 70% confluency with mimics or inhibitors. Following transfection, cells were treated with (A) paclitaxel (5nM) or (B) dasatinib (50nM). Cell death was quantified every 6 h for 24 h total using ImageJ and methylene blue dead cell staining. 19m, sdRNA-D19b mimic; 19i, sdRNA-D19b inhibitor; 24m, sdRNA-A24 mimic; 24i, sdRNA- A24 inhibitor; CTLm, scrambled mimic; CTLi, scrambled inhibitor; Mock, vehicle-treated control. * indicates P<0.001; P-values by unpaired two-tailed t-test as compared to Mock.
Figure 5. SdRNA-D19b and sdRNA-A24 mRNA targets. (A) Alignments between putative 3’UTR target sites with sdRNAs –D19b (top) and –A24 (bottom). Vertical lines indicate Watson-Crick basepair. Dotted lines indicate G:U basepair. TS1, target site 1. TS2, target site 2. (B) SdRNAs –D19b and –A24 specifically repress luciferase expression from mRNAs containing CD44 and CDK12 target sites in their 3’UTRs. SdRNA mimics and luciferase reporters with target sequences (bottom) and/or controls (LACTA refers to beta galactosidase control sequence) were constructed and cotransfected as previously described7,61. * indicates P<0.01; P-values by unpaired two-tailed t-test as compared to LACTA reporter transfected with the same mimic excepting siLACTA transfections comparing LACTA reporter to CD44 reporter.
Round 3
Reviewer 3 Report
I thank the authors for having stated how p-values were calculated in the figure legends.
Still I have the same concern regarding the drug treatment experiment. I believe that cells transfected with scrambled mimics/inhibitors are not the right control for transfected cells treated with the different drugs. To distinguish between the effect of the drug and a potential toxic effect of the transfection, the authors should show how vehicle-treated cells belonging from the same transfection behave. To be more clear, each drug treatment time point for 19m, 19i, 24m and 24i transfections has to be normalized to its corresponding vehicle-treated time point (19m vehicle 6h, 19m vehicle 12h, etc...). It is also not clear how many biological and/or technical replicates have been used.
I am sorry to say that for this reason I still think the manuscript is not yet ready to be accepted for publication.
Author Response
We appreciate the reviewer's continued diligence. While replicate number was included under methods, we agree its inclusion in the Figure 4 legend only adds to overall clarity and have revised the manuscript as such. That said, we suggest our last revision did adequately and directly address concern about transfection toxicity. Mock (vehicle-alone) treated cell viability at each time point was specifically included in both Figures 4A and 4B AND these values were directly used for comparison for determination of statistical significance. Importantly, viabilities (as compared to Mock) were not significantly affected by any of our transfections (19m, 19i, 24m, 24i, ctlM, or ctlI) excepting paclitaxel-treated 24M transfected cells (Figure 4A) and dasatinib -treated 19M transfected cells (Figure 4B) clearly arguing against appreciable toxic effects due to transfection itself.
Round 4
Reviewer 3 Report
I thank the authors for their reply and for adding the information required in the figure legend.
I respectfully disagree on the use of mock control as a readout of transfection toxicity and I am still convinced that the best control for the experiment is the one I suggested in the previous rounds of revision.
That said, I can convene that the effects observed for 19m transfection upon dasatinib treatment should not be caused by a transfection-toxicity side effect.
Therefore, I am leaving the decision whether to accept the manuscript in this form or to ask for additional controls to the editor keeping the review process open in the form of minor revision.